

# Enhancing privacy-preserving brain tumor classification with adaptive reputation-aware federated learning and homomorphic encryption

Swetha Ghanta[1], Prasanthi Boyapati[1], Sujit Biswas[2,3], Ashok K. Pradhan[1] and Saraju P. Mohanty[4]

[1] Department of Computer Science and Engineering, School of Engineering and Sciences, SRM University, AP, Guntur, Andhra Pradesh, India
[2] Computer Science Department, Northumbria University, Newcastle, United Kingdom
[3] Computer Science, University of London, London, United Kingdom
[4] Department of Computer Science and Engineering, University of North Texas, Denton, United States

Corresponding authors
Sujit Biswas,
sujit.biswas@northumbria.ac.uk
Ashok K. Pradhan,
ashokkumar.p@srmap.edu.in

## ABSTRACT

Brain tumor diagnosis using magnetic resonance imaging (MRI) scans is critical for improving patient survival rates. However, automating the analysis of these scans faces significant challenges, including data privacy concerns and the scarcity of large, diverse datasets. A potential solution is federated learning (FL), which enables cooperative model training among multiple organizations without requiring the sharing of raw data; however, it faces various challenges. To address these, we propose Federated Adaptive Reputation-aware aggregation with CKKS (Cheon-Kim-Kim-Song) Homomorphic encryption (FedARCH), a novel FL framework designed for a cross-silo scenario, where client weights are aggregated based on reputation scores derived from performance evaluations. Our framework incorporates a weighted aggregation method using these reputation scores to enhance the robustness of the global model. To address sudden changes in client performance, a smoothing factor is introduced, while a decay factor ensures that recent updates have a greater influence on the global model. These factors work together for dynamic performance management. Additionally, we address potential privacy risks from model inversion attacks by implementing a simplified and computationally efficient CKKS homomorphic encryption, which allows secure operations on encrypted data. With FedARCH, encrypted model weights of each client are multiplied by a plaintext reputation score for weighted aggregation. Since we are multiplying ciphertexts by plaintexts, instead of ciphertexts, the need for relinearization is eliminated, efficiently reducing the computational overhead. FedARCH achieved an accuracy of 99.39%, highlighting its potential in distinguishing between brain tumor classes. Several experiments were conducted by adding noise to the clients' data and varying the number of noisy clients. An accuracy of 94% was maintained even with 50% of noisy clients at a high noise level, while the standard FL approach accuracy dropped to 33%. Our results and the security analysis demonstrate the effectiveness of FedARCH in improving model accuracy, its robustness to noisy data, and its ability to ensure data privacy, making it a viable approach for medical image analysis in federated settings.

## INTRODUCTION

Brain tumors are a very critical condition, and immediate identification and treatment are required to improve patient survival rates. Diagnosis of brain tumors is often done using magnetic resonance imaging (MRI) and computed tomography (CT) scans. MRI scans are usually preferred over CT scans because they do not cause radiation exposure. Tumors can be of two types: benign and malignant. Malignant tumors are cancerous and require immediate treatment, while benign tumors are non-cancerous but necessitate frequent tests and patient monitoring.

Analyzing MRI scans is crucial in this context, but it is often time-consuming and requires expertise. Automating brain MRI image analysis presents several challenges. The major challenge is the availability of datasets; medical institutions often do not share their patient data to protect privacy. To address these challenges, a new paradigm called federated learning (FL) has emerged (*McMahan et al., 2017*). FL enables collaborative training of a global model across multiple clients by sharing only model updates, thereby preserving the privacy of raw data. In typical FL setups, a central server aggregates model weight updates from decentralized clients using algorithms such as federated averaging (FedAvg), thereby iteratively improving the global model. This paradigm has gained attention for enabling data-driven innovation while respecting data locality and legal constraints.

Although this approach seems to offer a solution, there are several issues associated with its real-time application. For example, these frameworks require a large amount of data, which is not always possible in the medical domain, as some medical conditions can be extremely rare and underrepresented. To address this problem, we utilize transfer learning (TL). By employing pre-trained models, we can leverage existing knowledge and adapt it to our specific problem with limited data. This approach helps mitigate the challenge of data scarcity by fine-tuning models on small, specialized datasets, thereby improving performance even when the amount of local data is limited (*Khan et al., 2022b*).

In FL, the global model is trained using the weights received from the clients. However, if a client sends malicious or erroneous data to the central server, which treats all clients equally, the global model will use these erroneous updates for aggregation. This can eventually corrupt the global model and affect all clients. Several works (*Fan et al., 2023*; *Kang & Ahn, 2023*) have been proposed to address this problem, but most are based on a cross-device scenario rather than a cross-silo scenario. A cross-device scenario involves IoT devices, where the number of devices is large but their computational ability is limited. In contrast, a cross-silo scenario involves organizations where the number of entities is smaller but their computational ability is higher. For our use case, we consider a cross-silo

scenario where multiple medical institutions collaborate for FL. In a cross-device scenario, existing solutions often reject the weights from underperforming clients and only consider the weights from the best-performing clients. This approach is feasible in a cross-device scenario because the server can choose from a large pool of clients. However, in a cross-silo scenario, where the number of clients is already limited, completely rejecting a client's update can increase bias towards certain clients, ignoring others.

To address these issues, we propose FedARCH, a novel framework where reputations are assigned to each client based on their performance evaluation. Instead of using a simple FedAvg approach, where all model weights are aggregated using a simple average, a reputation-based weighted aggregation is employed. This process is iterated after each round of training, as client performance and, therefore, reputations can change after any round. To prevent sudden changes in client performance from unduly affecting the assigned reputations, we have implemented a smoothing factor. This factor stabilizes the reputation adjustments, preventing abrupt increases or decreases from causing significant fluctuations. Additionally, as the training progresses across multiple rounds, more recent performance updates must have a greater influence on the reputation, while older updates should gradually diminish in impact. To achieve this, we incorporate a decay factor that reduces the weight of older reputations, allowing the system to adapt to the recent client performances. We will discuss these details in the upcoming sections.

Another potential issue in FL is the model inversion attack (*Fredrikson, Jha & Ristenpart, 2015*), where a malicious actor can reconstruct the original data from the shared weights, thus compromising privacy. To address this problem, researchers developed homomorphic encryption (HE), which allows aggregation to be performed on encrypted data without decrypting it. In FedARCH, we used Cheon-Kim-Kim-Song (CKKS) HE (*Cheon et al., 2017*), a somewhat homomorphic encryption scheme (SHE). We have specifically chosen CKKS over other HE schemes like RSA and Paillier, because:

- CKKS is based on the hardness of ring learning with errors (RLWE) problem, which is considered to be quantum-resistant, offering security against potential quantum attacks while enabling efficient encrypted computations.
- CKKS allows a limited number of both addition and multiplication operations on encrypted data, which is necessary for our weighted aggregation, unlike other HE schemes that support only one of the two operations.

Some of the other popular RLWE-based HE schemes include Brakerski/Fan-Vercauteren (BFV) and Brakerski-Gentry-Vaikuntanathan (BGV) (*Fan & Vercauteren, 2012*; *Brakerski, Gentry & Vaikuntanathan, 2014*). However, CKKS HE was selected because it operates on approximations, significantly enhancing computational efficiency. CKKS can handle real numbers, enabling it to support the complex arithmetic required for our model. This approximate arithmetic capability makes CKKS faster compared to other schemes that operate on exact arithmetic, providing a good balance between security and performance for our proposed FedARCH framework.

# CONTRIBUTIONS OF THE CURRENT PAPER

## Motivation

Most existing FL research focuses on cross-device scenarios, which involve numerous simple Internet of Things (IoT) devices or mobile phones with limited computational capabilities and intermittent connectivity. These studies typically assume high dropout rates, ignore underperforming clients, and don't provide feedback on client contributions. While these assumptions may suit cross-device FL, they do not apply to cross-silo FL, where multiple organizations, such as medical institutions, collaborate with substantial, valuable data.

In contrast to cross-device FL, the stakes are notably higher for cross-silo FL. Here, each client represents an organization, contributing critical and sensitive data, especially relevant in domains like healthcare. Ignoring any client, even an underperforming one, risks losing essential data. Organizations in this setting are generally more reliable and experience lower dropout rates than individual devices, making it essential to devise sophisticated approaches to handling client contributions effectively. Furthermore, providing performance feedback to clients in cross-silo FL can help organizations understand their contributions' impact on the global model. Such feedback enables institutions to improve their local models and strengthen future contributions to the global model.

## Problem addressed

FL applications in medical image analysis face multiple challenges that limit their potential. Key issues include untrusted third-party servers, inadequate client data validation, calculating accurate client reputations, and managing dynamic performance variations. Many existing solutions only address one or a few of these challenges, often at the cost of overall system performance and increased computational overhead. For FL to be fully effective in sensitive fields like healthcare, these challenges must be addressed in a unified manner without sacrificing model performance.

## Solution proposed

We propose FedARCH, a novel FL framework that evaluates each client's contribution before incorporating it into the global model, using an adaptive reputation mechanism with smoothing and decay factors to maintain dynamic, reliable reputations. This adaptive reputation mechanism factors in both recent and historical performance, ensuring that contributions remain meaningful over time while mitigating the influence of sudden performance changes.

To address the challenge of the untrusted server, we employ the CKKS HE technique, which enables secure operations on encrypted weights, thereby protecting the data from model inversion attacks. CKKS is particularly well-suited as it supports both addition and multiplication operations on real numbers, a feature that other HE schemes lack. This setup allows the server to work exclusively with encrypted data without needing decryption, maintaining data privacy. The computational overhead associated with using CKKS HE is reduced by using the plaintext-ciphertext multiplications instead of

ciphertext-ciphertext multiplications. This greatly reduces the ciphertext growth and noise accumulation.

### Novelty and significance of the solution

FL holds tremendous potential to automate medical image analysis, yet its benefits in critical fields are hindered by ongoing security and performance challenges. FedARCH addresses these issues comprehensively without compromising model accuracy.

The primary contributions of this work include:

1. FedARCH, an innovative cross-silo FL framework—Featuring adaptive reputation-based weighted aggregation for real-time performance management, particularly useful in classifying brain tumors from MRI scans.
2. Client performance evaluation—Using validation reports from neighboring clients, the system provides feedback to underperforming clients, encouraging continuous improvement.
3. Incorporation of optimized CKKS HE—This approach effectively guards against model inversion attacks from an untrusted server without compromising on computational efficiency.
4. Extensive simulations with variable client performance—Compared with the standard FL algorithm, FedARCH demonstrates superior performance, especially in scenarios with multiple underperforming clients.

The proposed FedARCH framework advances the field by enhancing both security and model performance, particularly in high-stakes applications like medical imaging.

## RELATED PRIOR RESEARCH

With the advent of deep learning (DL) and convolutional neural networks (CNNs), several research articles have been published to address the problem of brain tumor classification using DL techniques. A 23-layer CNN model was proposed for brain tumor classification on the Figshare dataset, while TL was also applied to address a binary classification task on a smaller Harvard dataset (*Khan et al., 2022b*). To further leverage TL, an ensemble approach was employed for feature extraction across multiple TL models, combining the top three models using a multi-layer perceptron (MLP) (*Remzan et al., 2024*). For the same classification problem, YOLOv7 was utilized, incorporating a convolutional block attention module (CBAM) to enhance feature extraction (*Abdusalomov, Mukhiddinov & Whangbo, 2023*).

Although these approaches generate high-performing accuracies, they are based on simple local learning models trained on smaller datasets, which may lack generalizability when applied to different datasets. Centralized learning (CL), where all data is collected and processed at a single location, poses additional challenges, including the risk of a single point of failure and reluctance from organizations to participate due to concerns about patient data privacy. To address these issues, researchers introduced FL, a collaborative learning technique that preserves patient privacy by working with model updates rather than raw data.

FL has gained significant attention as an approach to train models across decentralized devices or institutions while preserving data privacy. Initially, the FedAvg algorithm was introduced, enabling local models to be trained independently on each client and subsequently aggregated using a simple average to form a global model that synthesizes knowledge from all clients (*McMahan et al., 2017*). Building on this foundation, FL was first applied to medical image analysis, demonstrating its potential in sensitive domains (*Sheller et al., 2020*). To further enhance FL's performance, ensemble and voting techniques were integrated to improve classification accuracy in complex datasets (*Islam et al., 2023*). Additionally, TL techniques were combined with FL specifically for brain tumor classification, allowing the model to be evaluated across various client contribution levels and performance metrics, thus highlighting the adaptability of FL in handling diverse data distributions (*Viet et al., 2023*). The importance of leveraging multimodal data in FL has been widely recognized, with recent work discussing the key challenges associated with integrating heterogeneous data sources within FL frameworks (*Huang et al., 2024*).

While effective, model inversion attacks (*Fredrikson, Jha & Ristenpart, 2015*) pose a significant threat to FL systems by reconstructing sensitive data from model updates. Various defense mechanisms have been considered, including differential privacy (*Dwork & Roth, 2014*) and secure multi-party computation (*Zhao et al., 2019*), but these often come with trade-offs in terms of computational overhead and model accuracy. To address these challenges and preserve data privacy, secure aggregation techniques were explored to ensure that the central server cannot access individual model updates (*Bonawitz et al., 2017*). Recent advancements, such as the use of HE (*Cheon et al., 2017*), enable computation on encrypted data, eliminating the need for decryption in a zero-trust architecture and further enhancing privacy in FL systems. The SHE approach was employed for cancer image analysis, incorporating an additive secret sharing technique (*Truhn et al., 2024*). But since all clients are treated equally and their updates are aggregated to update the global model, ignoring the issue of underperforming clients can affect the performance of the final model.

To address client contribution disparity, weighted aggregation was utilized based on a data quality factor, along with the EL-Gamal HE technique (*Zhang et al., 2022*). Since EL-Gamal supports multiplicative homomorphism, the encryption scheme was modified to enable additive homomorphism, thereby reducing communication overhead. The FedRaHa framework was proposed (*Panigrahi, Bharti & Sharma, 2023*), incorporating reputations for client selection based on cosine similarity scores and employing hierarchical aggregation to reduce communication overhead. A lightweight privacy-preserving federated learning (LPBFL) scheme was introduced to calculate the reputation of each client prior to aggregating their updates into the global model, thereby preventing malicious updates from poisoning the final model. Their work utilized Paillier HE to maintain the privacy of local model updates (*Fan et al., 2023*). However, Paillier is a partial HE scheme, which supports only either addition or multiplication operations, and it is considerably slow. A genetic algorithm (GA) approach was proposed to optimize client

**Table 1 Summary of prior works in brain tumor classification.**

| Reference | Dataset | Approach | Accuracy |
|---|---|---|---|
| *Khan et al. (2022b)* | Figshare | 23-layer CNN | 97.8% |
| *Mathivanan et al. (2024)* | Kaggle | MobileNetV3 | 99.75% |
| *Rasool et al. (2022)* | Kaggle | GoogleNet-SVM | 98.01% |
| *Senan et al. (2022)* | SARTAJ | AlexNet-SVM | 95.10% |
| *Khan et al. (2022a)* | Kaggle | Hierarchical deep learning-based brain tumor (HDL2BT) classification | 92.13% |
| *Lamrani et al. (2022)* | Kaggle | CNN model for binary classification | 96.33% |
| *Gaur et al. (2022)* | SARTAJ | CNN and explainable AI | 94.64% |
| *Vidyarthi et al. (2022)* | Kaggle | NN classifier with cumulative variance method (CVM) for feature selection | 95.86% |
| *Albalawi et al. (2024)* | Kaggle | VGG with FL | 98% |
| *Islam et al. (2023)* | Kaggle | Voting ensemble of 6 TL models | With FL 91.05% |
| | | | Without FL 96.68% |
| *Viet et al. (2023)* | Figshare | VGG with FL | 98.69% |
| *Ay, Ekinci & Garip (2024)* | – | FedAvg | 85.55% |
| *Zhou, Wang & Zhou (2024)* | SARTAJ | FL with EfficientNetB0 and ResNet50 | 80.17% |
| | | | 65.32% |

selection for FL with minimizing the communication cost as the objective function (*Kang & Ahn, 2023*). But, the use of genetic algorithm can significantly increase the training time and is not suitable for larger datasets and a huge number of clients. A private blockchain-based framework was considered for storing model weights in chunks rather than directly, with miners tasked with evaluating the quality of local updates (*Bhatia & Samet, 2023*). The major limitation of their work is the scalability; if the number of clients increases, then the number of weights will increase predominantly, thus making the idea of storing the weights in blockchain inefficient. Table 1 summarizes various existing works in the field of brain tumor classification tasks. In summary, while significant progress has been made in addressing data privacy, model robustness, and client heterogeneity in FL. However, there are still some challenges, particularly in cross-silo scenarios. Hence, FedARCH builds on these foundations by introducing reputation-based weighted aggregation, smoothing, and decay factors for dynamic performance management, and the integration of CKKS HE to enhance privacy and security. CKKS HE, in particular, is notable for its efficient handling of approximate arithmetic, making it especially suitable for FL applications.

## PRELIMINARIES

### Federated learning

FL is a latest trending paradigm in the machine learning (ML) community, offering solutions to several problems such as data scarcity, data privacy, and real-time collaborative learning. FL has gained significant accolades for its capability to allow multiple parties to collaborate and train a global model without sharing their raw data; rather, they share weight updates. This replaced the CL scenario, where data from multiple clients is collected in a cloud server and is used to train a global model, which also resides

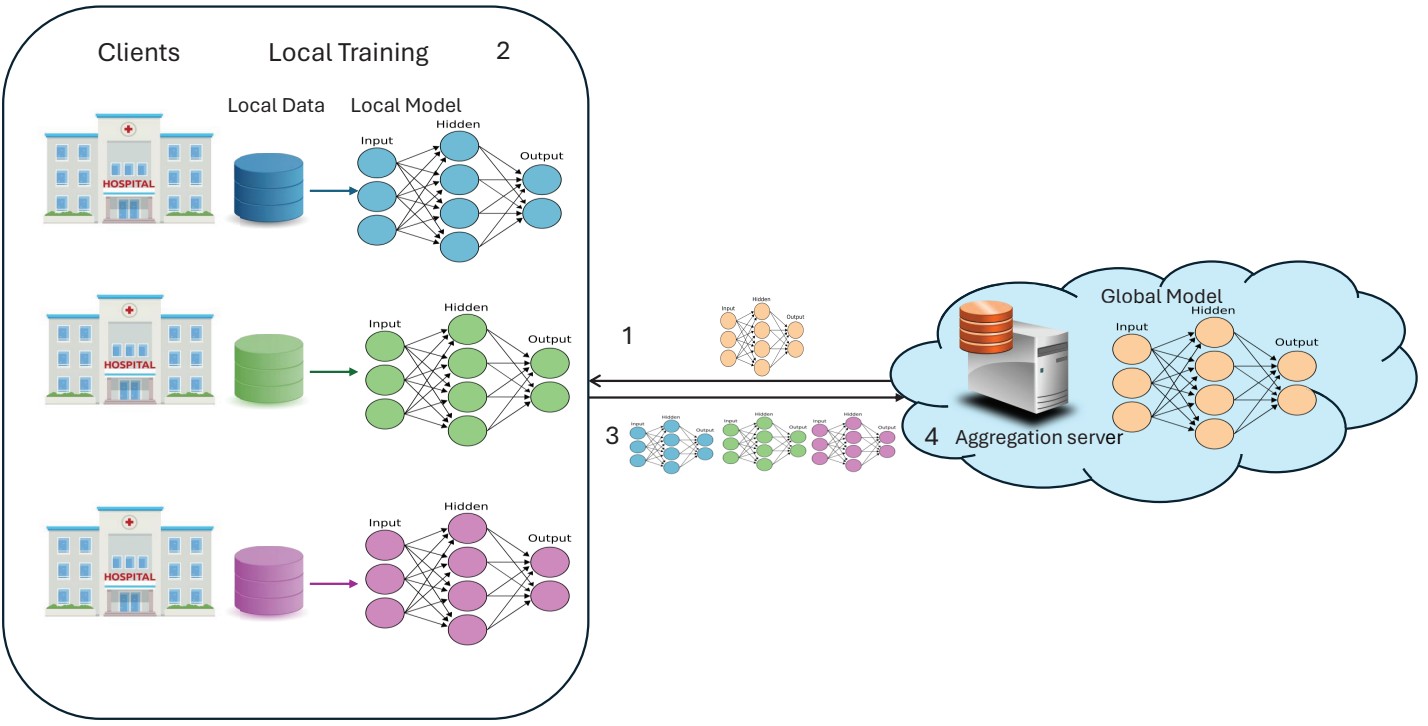

**Figure 1 A simple FL architecture.**

in the same cloud. FL implementation requires the following components and is illustrated in Fig. 1:

- Server: A server is a cloud environment that holds the global model and acts as an aggregation server, aggregating the weights from clients.
- Client: A client can be any organization or medical institution in a cross-silo scenario, while in a cross-device scenario, it can be any device, like mobiles, IoT devices, *etc.*
- Global model: In FL, multiple clients collaboratively train a global model, which can be any ML or DL model.
- Local model: Each client receives a replica of the global model to train on its local data, and at the client side, it is referred to as the local model.
- Model weights: When training the local model with local data, model weights are obtained. These weights represent the learned parameters of the neural network, determining the importance of input features, controlling the strength of neural connections, and encapsulating the model's knowledge gained from the training data.

The typical workflow of FL involves the server distributing the initial global model to all clients. Each client trains the model on its local data, updates the model weights, and sends these updates back to the server. The server aggregates the updates to form a new global model, which is then redistributed to the clients. This process repeats for a predefined number of rounds or until convergence.

## CKKS homomorphic encryption

To further enhance data privacy and security in FL, especially when dealing with a curious server that might attempt to infer sensitive information from model updates, we employ CKKS HE (*Cheon et al., 2017*). CKKS HE is a type of somewhat homomorphic encryption scheme that supports arithmetic operations on encrypted data without needing to decrypt it, ensuring that data remains secure during the aggregation process. Key components of CKKS HE within FL include:

- CKKS context: This holds parameters such as the polynomial modulus degree, scaling factor, security parameters, and the public-secret key pair. It defines the encryption scheme's environment, setting up the structure for encryption, decryption, and homomorphic operations.
- Message encoding and decoding:
  Encoding: In CKKS, real numbers are encoded into a polynomial ring to enable encrypted operations (*Huynh, 2020*). The message $m$ is transformed into a plaintext polynomial $\Delta m(x)$, where $\Delta$ is a scaling factor used to maintain precision during homomorphic operations by converting floating-point numbers to large integers. This is done by multiplying the floating-point numbers by the scaling factor before encryption, enabling accurate representation within the encryption scheme. This encoding maps the message into the ring.

$$R = Z[x]/(x^N + 1) \tag{1}$$

where $Z$ represents integers, $(x^N + 1)$ is a cyclotomic polynomial, and $N$ is the polynomial degree, usually represented as powers of 2:

$$m \to \Delta m(x) \in Z[x]/(x^N + 1). \tag{2}$$

This polynomial representation allows CKKS to perform homomorphic operations like addition and multiplication on encrypted data, with the operations corresponding to similar operations on plaintext polynomials.

Decoding: The reverse process that maps the polynomial back to real numbers.

- Key generation: Generate public and private keys: $(pk, sk)$, where $pk$ is used for encryption and $sk$ for decryption. Each plaintext polynomial is encrypted using a public key, resulting in ciphertexts of the form: $c_1 = (c_{1,0}, c_{1,1})$ and $c_2 = (c_{2,0}, c_{2,1})$, where $c_{i,j}$ is a polynomial in $Z_q[x]/(x^N + 1)$
- Encryption: Given a plaintext polynomial $m(x)$, the encryption function using public key $pk = (a, b)$ and a random noise $e$ generates a ciphertext $c$, a pair of polynomials where,

$$c = Enc(m(x), pk) = (c_0, c_1) \tag{3}$$

$$c_0 = b.s + m + e_0 \tag{4}$$

$$c_1 = a + e_1. \tag{5}$$

- Homomorphic operations:

Both addition and multiplication operations are performed on the ciphertexts, producing encrypted results that approximate the arithmetic operations on the underlying plaintexts.

Addition:

$$Enc(m_1(x), pk) + Enc(m_2(x), pk) = Enc(m_1(x) + m_2(x), pk)$$
$$= (c_{1,0} + c_{2,0}, c_{1,1} + c_{1,2}) \tag{6}$$

where $(c_{1,0}, c_{1,1})$ and $(c_{2,0}, c_{2,1})$ are ciphertexts for $m_1(x)$ and $m_2(x)$ respectively.

Multiplication:

When two ciphertexts are multiplied, it is not as straightforward as addition; the polynomial representations of ciphertexts expand:

$$c_{mul} = c_1 * c_2 \tag{7}$$

since each ciphertext is a tuple $(c_0, c_1)$, the multiplication expands as follows:

$$c_{mul} = (c_{1,0}, c_{1,1}) * (c_{2,0}, c_{2,1}) = (c_{1,0}c_{2,0}, c_{1,0}c_{2,1} + c_{1,1}c_{2,0}, c_{1,1}c_{2,1}). \tag{8}$$

This results in a new third-term ciphertext, *i.e.*,

$$c_{mul} = (c'_0, c'_1, c'_2). \tag{9}$$

Ciphertext multiplication increases the size of the ciphertext—initially from two components to three, then five, nine, and so on. To prevent this uncontrolled growth, relinearization is applied to reduce the ciphertext back to the standard 2-component format and maintain its size. However, relinearization introduces additional computational complexity and overhead.

- Decryption: Given a ciphertext $c$, the decryption function returns the plaintext polynomial

$$m(x) = Dec(c, sk). \tag{10}$$

In the FedARCH framework, encrypted model weights $(E_i^t)$ of the client $i$ at round $t$ are multiplied by a plaintext normalized reputation score $(\bar{R}_i^t)$ for weighted aggregation. Since we multiply ciphertexts by plaintexts, rather than by other ciphertexts, relinearization is not required. Relinearization, typically used in HE schemes, manages the growth of ciphertext size and complexity after multiplying ciphertexts together. By avoiding the need for relinearization, we simplify our computational process and reduce overhead. These weights from different clients are further added together using ciphertext addition, which is a straightforward operation in CKKS.

Integrating CKKS HE into our FL framework provides a robust solution to protect sensitive client data from potential privacy breaches by the central server. This approach ensures that even if the server is compromised or curious, it cannot access or infer the original data, thus maintaining the confidentiality and privacy of each client's data throughout the training process.

## FEDARCH FRAMEWORK

We propose FedARCH, a novel FL framework for collaborative learning in a cross-silo scenario. In this framework, we created a simulated environment with 10 clients, where each client represents a medical institution. A central server, referred to as the aggregation server, holds the global model used for the FL process. The server performs the aggregation

of client weights after each FL round, and this process is repeated until the specified number of rounds is reached.

In this scenario, we assume the server is not trustworthy and it could perform a model inversion attack to obtain the original data, thus being termed as a "curious" server. We also assume that clients are trustworthy, meaning they do not perform a model inversion attack or intentionally send malicious or erroneous updates. However, clients can still underperform due to several reasons:

Data heterogeneity: Clients have different data distributions. For example, medical institutions may have varying case severities, leading to differences in model performance.

Resource Constraints: Some clients might have limited computational resources, resulting in less effective training.

Model training issues: Suboptimal hyperparameter settings, insufficient training epochs, or software bugs can cause variations in local model performance.

Environmental factors: Factors like network latency or power outages could impact the training process for some clients.

Data quality: Variations in data quality across clients, such as noisier or less representative data, can lead to poorer model performance.

By considering these factors, the FedARCH framework aims to accommodate and adjust for underperforming clients through reputation-based weighted aggregation, smoothing, and decay factors, ensuring that the global model remains robust and accurate despite these variations. The working of the proposed FedARCH framework is presented in detail through Algorithms 1 to 5.

Each client trains the global model, enhancing decision-making by participating in the FL process with local data while ensuring data privacy by sharing only the model weights. In the FedARCH framework, we use the pre-trained ResNet18 model (*He et al., 2016*) and fine-tune it for our specific use case. A replica of the global model $W^0$ is shared with all clients. Upon receiving the model, each client $C_i$ trains it with their local data $\mathscr{D}_{train}$. To preserve privacy from a curious server, the local model weights $W_i^t$ generated at each client are encrypted using CKKS HE to obtain the encrypted local model weights $E_i^t$. Figure 2 provides an overview, and Fig. 3 a detailed illustration of the proposed FedARCH framework.

To ensure synchronization and traceability, model updates are versioned as $E_i^t$, where $i$ denotes the client index and $t$ represents the round number. These version identifiers help clients and server consistently validate the correct model state and enable efficient tracking throughout the process. Each client $C_i$ shares its $E_i^t$ with the server for aggregation and with the next client $C_{i+1}$ for validation. In this framework, each client $C_i$ also acts as a validator for its previous client $C_{i-1}$ within a pre-defined time window.

Specifically, client $C_i$ validates the local weights $E_{i-1}^t$ of the previous client $C_{i-1}$ using the validation data $\mathscr{D}_{val}$ and generates a validation score (val_score) $P_{prev}^t$ for that previous client, which is then sent to the server. The next client, upon receiving the previous client's encrypted local model weights $E_{prev}^t$, decrypts them using CKKS decryption to obtain the

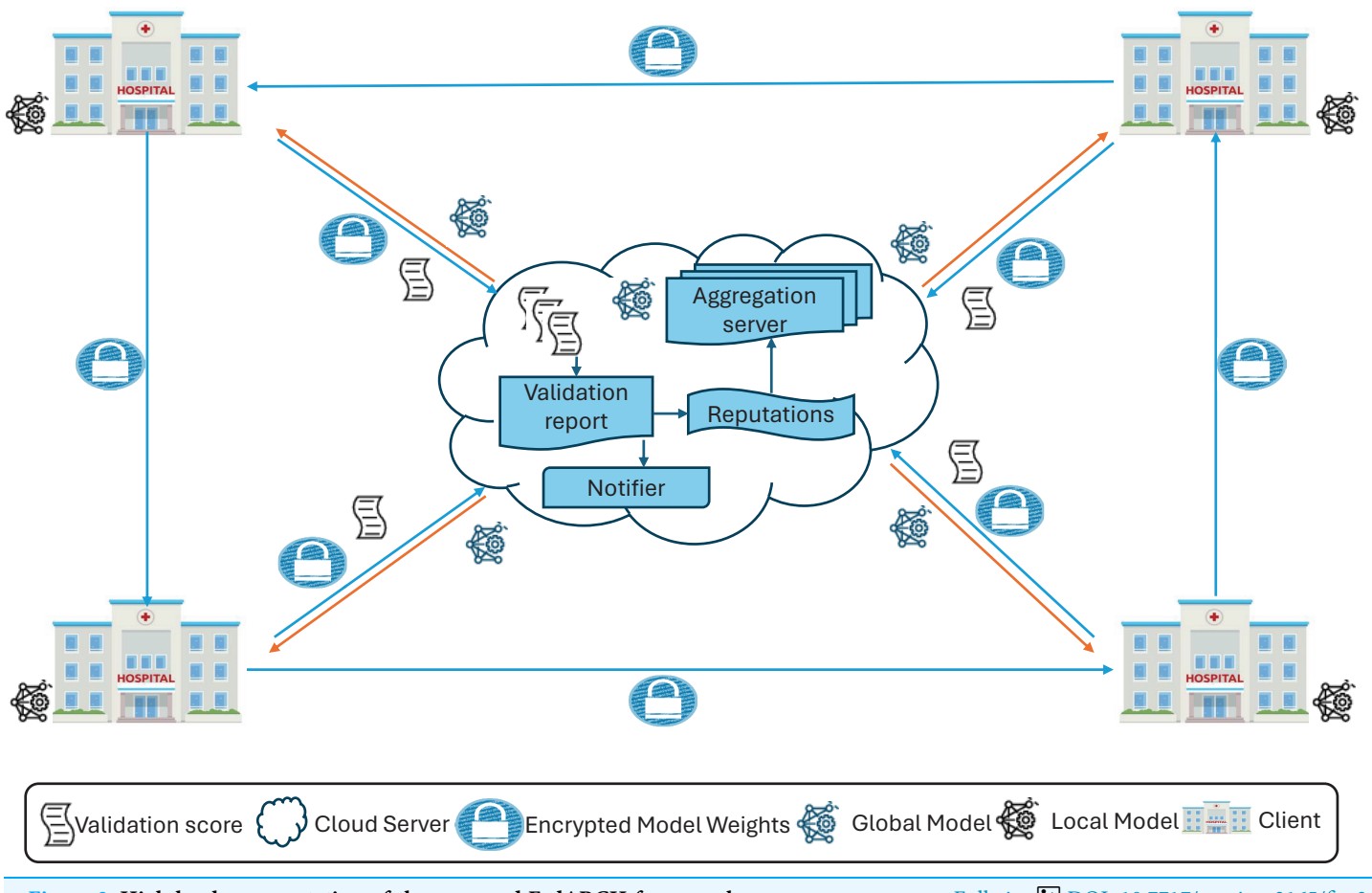

**Figure 2  High-level representation of the proposed FedARCH framework.**

local model weights $W_{prev}^t$. To facilitate this, we assume that all clients share a common CKKS encryption *context* with an implicit public-private key pair managed by a trusted Certificate Authority (CA). This ensures that each client can securely decrypt the weights from the previous client using the shared *context*. This validation mechanism provides an additional layer of accountability and accuracy, reducing potential biases and ensuring a more comprehensive evaluation of the model's performance across various datasets.

At the server side, communication is synchronous within each round: the server waits until the time window concludes to collect all val_scores from the client validators before proceeding to the next aggregation step. This design ensures that client feedback is aligned temporally and version-wise, addressing synchronization concerns and reducing inconsistencies due to delayed or missing validations.

Upon receiving the val_scores from all clients, the server's notifier informs underperforming clients if their validation score falls below a threshold value, defined as the average of the validation accuracies of all clients in that round. This notification helps clients take appropriate measures to improve their local data or training processes. Although clients could validate themselves, the notifier is necessary because clients do not have access to the validation accuracies of other clients to calculate this threshold. As a part

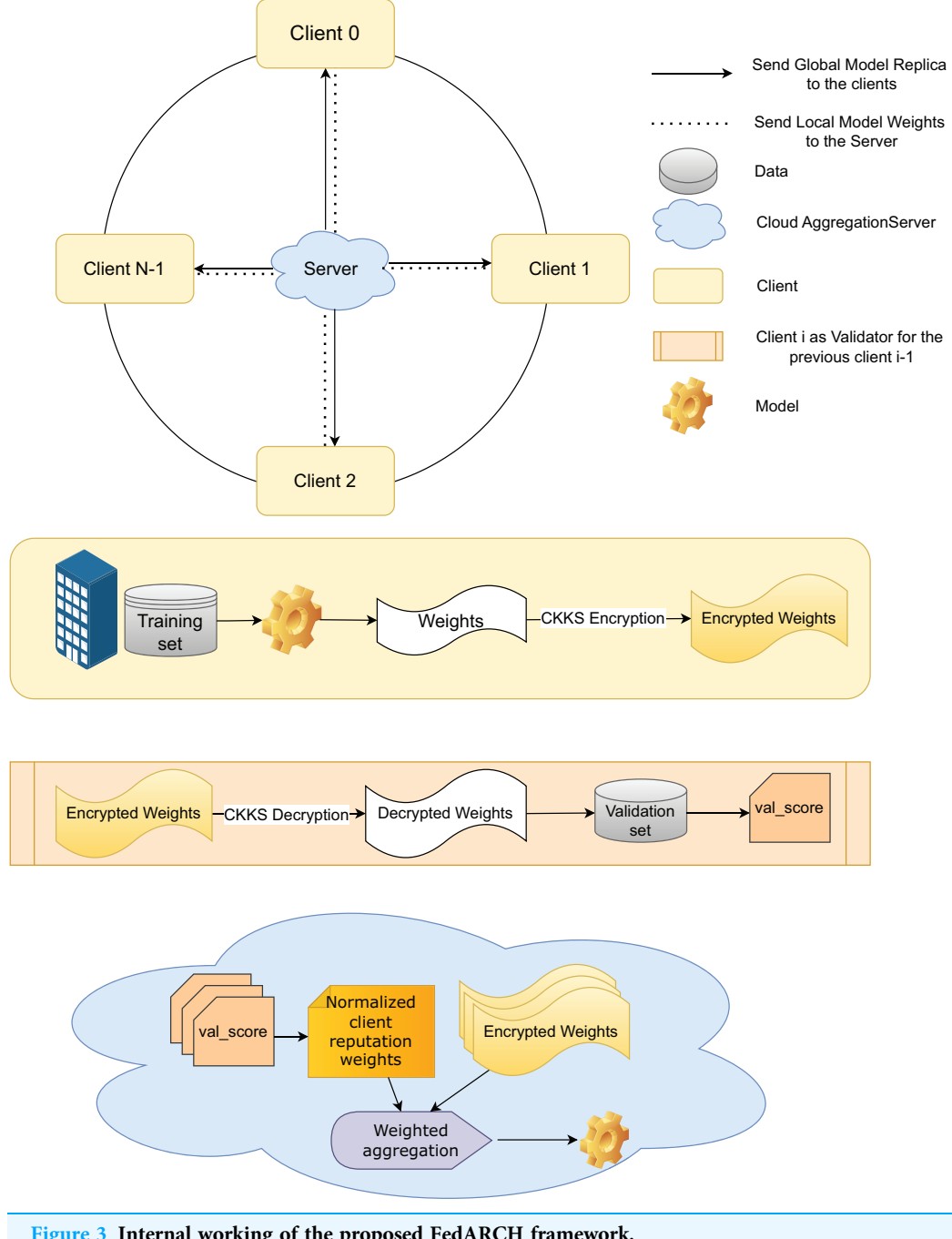

**Figure 3  Internal working of the proposed FedARCH framework.**

of the server, the notifier ensures that clients receive the necessary feedback to enhance their performance. The server then assigns a reputation value $R_i^t$ to each client using the val_scores. These reputations are updated and adjusted using smoothing and decay factors. A smoothing factor $\alpha$ is employed to handle sudden increases or decreases in client performance and maintain stability, while a decay factor $\beta$ reduces the impact of older

---

**Algorithm 1**   **Federated learning with reputation and CKKS encryption.**

**Require:** Training dataset $\mathcal{D}_{train}$, Validation dataset $\mathcal{D}_{val}$, Testing dataset $\mathcal{D}_{test}$, Set of clients $\mathcal{C}$, Number of clients $N$, Number of rounds $R$, Smoothing factor $\alpha$, Decay factor $\beta$, CKKS context *context*

**Ensure:** Final global model $W^R$

1: Split $\mathcal{D}_{train}$ among $N$ clients
2: Initialize global model $W^0$
3: Initialize reputations $R_i^0 \leftarrow 1$ for all clients $i$
4: Distribute $\mathcal{D}_{val}$ to all $N$ clients
5: **for** $t = 0$ to $R - 1$ **do**
6:     **for** each client $C_i \in \mathcal{C}$ **in parallel do**
7:         Train local model and obtain $W_i^t$
8:         $E_i^t \leftarrow \text{CKKSEncryption}(W_i^t, context)$
9:         Send $E_i^t$ to client $(i + 1) \mod N$
10:     **end for**
11:     **for** each client $C_i \in \mathcal{C}$ **in parallel do**
12:         Call VALIDATION($E_{(i-1+N) \mod N}^t$, context)
13:     **end for**
14:     Collect all $E_i^t$ and validation scores $P_i^t$ at the global server
15:     Call UPDATEREPUTATION($P_i^t, R_i^t, \alpha, \beta$)
16:     **Update global model weights:**
17:     $E^{t+1} = \sum_{i=1}^{N} \bar{R}_i^{t+1} \cdot E_i^t$         (Aggregate weights)
18:     **for** each client $C_i \in \mathcal{C}$ **in parallel do**
19:         $W^{t+1} \leftarrow \text{CKKSDecryption}(E^{t+1}, context)$
20:     **end for**
21: **end for**
22: $W^R \leftarrow W^{t+1}$
23: Evaluate final global model $W^R$ on $\mathcal{D}_{test}$

---

reputations, ensuring the model adapts to the latest updates. The working of the smoothing and decay factors is given by Eq. (11), and the notations are described in Table 2.

$$R_i^{t+1} = (\alpha \cdot R_i^t + (1 - \alpha) \cdot P_i^t) \cdot \beta. \tag{11}$$

If the smoothing factor $\alpha$ is high (closer to 1), the new reputation will rely more heavily on the previous reputation, reducing the influence of the current performance. This makes the system less sensitive to sudden changes or fluctuations in client performance. On the other hand, if $\alpha$ is low (closer to 0), the current performance will have a greater influence, making the reputation more responsive to recent client behavior. For the decay factor, if $\beta$ is close to 1, the reputations will maintain their value over time, retaining a strong memory of both past and current performance. If $\beta$ is closer to 0, the reputations will gradually

---

**Algorithm 2** **Validation.**

**Require:** Encrypted weights $E_{prev}^t$, CKKS context *context*
**Ensure:** Validation score $P_{prev}^t$
1: $W_{prev}^t \leftarrow$ CKKSDecryption($E_{prev}^t$, *context*)
2: Validate the model using $\mathscr{D}_{val}$
3: Store validation score $P_{prev}^t$

---

**Algorithm 3** **UpdateReputation.**

**Require:** Validation scores $P_i^t$, Reputations $R_i^t$, Smoothing factor $\alpha$, Decay factor $\beta$
**Ensure:** Updated and normalized reputations $\bar{R}_i^{t+1}$
1: **for** each client $i$ **do**
2: $\quad R_i^{t+1} = (\alpha \cdot R_i^t + (1-\alpha) \cdot P_i^t) \cdot \beta$
3: **end for**
4: Normalize reputations $\bar{R}_i^{t+1} = \frac{R_i^{t+1}}{\sum_{j=1}^{N} R_j^{t+1}}$

---

**Algorithm 4** **CKKS encryption.**

**Require:** Local model weights $W_i$, CKKS context *context*
**Ensure:** Encrypted local model weights $E_i$
1: $E_i \leftarrow \{\}$
2: **for** each layer $k$ in $W_i$ **do**
3: $\quad vector \leftarrow$ Flatten($W_i[k]$)
4: $\quad E_i[k] \leftarrow$ CKKSEncrypt($vector$, *context*)
5: **end for**
6: **return** $E_i$

---

**Algorithm 5** **CKKS decryption.**

**Require:** Encrypted local model weights $E_i$, CKKS context *context*
**Ensure:** Decrypted local model weights $W_i$
1: $W_i \leftarrow \{\}$
2: **for** each layer $k$ in $E_i$ **do**
3: $\quad decrypted\_vector \leftarrow$ CKKSDecrypt($E_i[k]$, *context*)
4: $\quad W_i[k] \leftarrow$ Reshape($decrypted\_vector$)
5: **end for**
6: **return** $W_i$

---

decay, allowing newer updates to have a stronger influence while older updates lose significance. The choice of these factors can be dynamically adjusted by the server based on the validation scores obtained from the clients.

**Table 2 Notations used in federated learning with reputation and CKKS encryption.**

| Notation | Description |
|---|---|
| $\mathscr{D}_{train}$ | Training dataset |
| $\mathscr{D}_{val}$ | Validation dataset |
| $\mathscr{D}_{test}$ | Testing dataset |
| $\mathscr{C}$ | Set of clients |
| $N$ | No. of clients |
| $R$ | No. of rounds |
| $\alpha$ | Smoothing factor for reputation update |
| $\beta$ | Decay factor for reputation update |
| $W^0$ | Initial global model weights |
| $W^t$ | Global model weights at round $t$ |
| $W_i^t$ | Local model weights of client $i$ at round $t$ |
| $E_i^t$ | Encrypted local model weights of client $i$ at round $t$ |
| $R_i^t$ | Reputation of client $i$ at round $t$ |
| $\bar{R}_i^t$ | Normalized reputation of client $i$ at round $t$ |
| $P_i^t$ | Validation score of client $i$ at round $t$ |
| $P_{prev}^t$ | Validation score of the previous client at round $t$ |
| $E_{prev}^t$ | Encrypted local model weights of the previous client at round $t$ |
| $W_{prev}^t$ | Local model weights of the previous client at round $t$ |
| $context$ | CKKS encryption context |

The reputations are then normalized to obtain the normalized reputation weight score $\bar{R}_i^t$ for each client. Using these plaintext normalized reputation weights, the server performs weighted aggregation on the clients' encrypted local model weights, optimizing CKKS HE to perform addition and multiplication operations on encrypted data without increasing the computational complexity. This process is represented in Eq. (12).

$$E^{t+1} = \sum_{i=1}^{N} \bar{R}_i^{t+1} \cdot E_i^t. \tag{12}$$

After the weighted aggregation, the initial global model $W_0$ is updated with the new aggregated weights $W_t$, which are then sent to all clients to update their local models. These aggregated weights remain in encrypted form, so the clients decrypt them using CKKS decryption before updating their local models. This entire process is repeated for $R$ rounds or until convergence. After $R$ rounds, the final global model $W^R$ is evaluated on the test data $\mathscr{D}_{test}$ using various evaluation metrics.

## EXPERIMENTAL RESULTS

### Dataset

For implementing FedARCH, we have considered the Kaggle dataset (*Nickparvar, 2021*) containing 7,023 brain MRI images with four class labels: meningioma, glioma, pituitary,

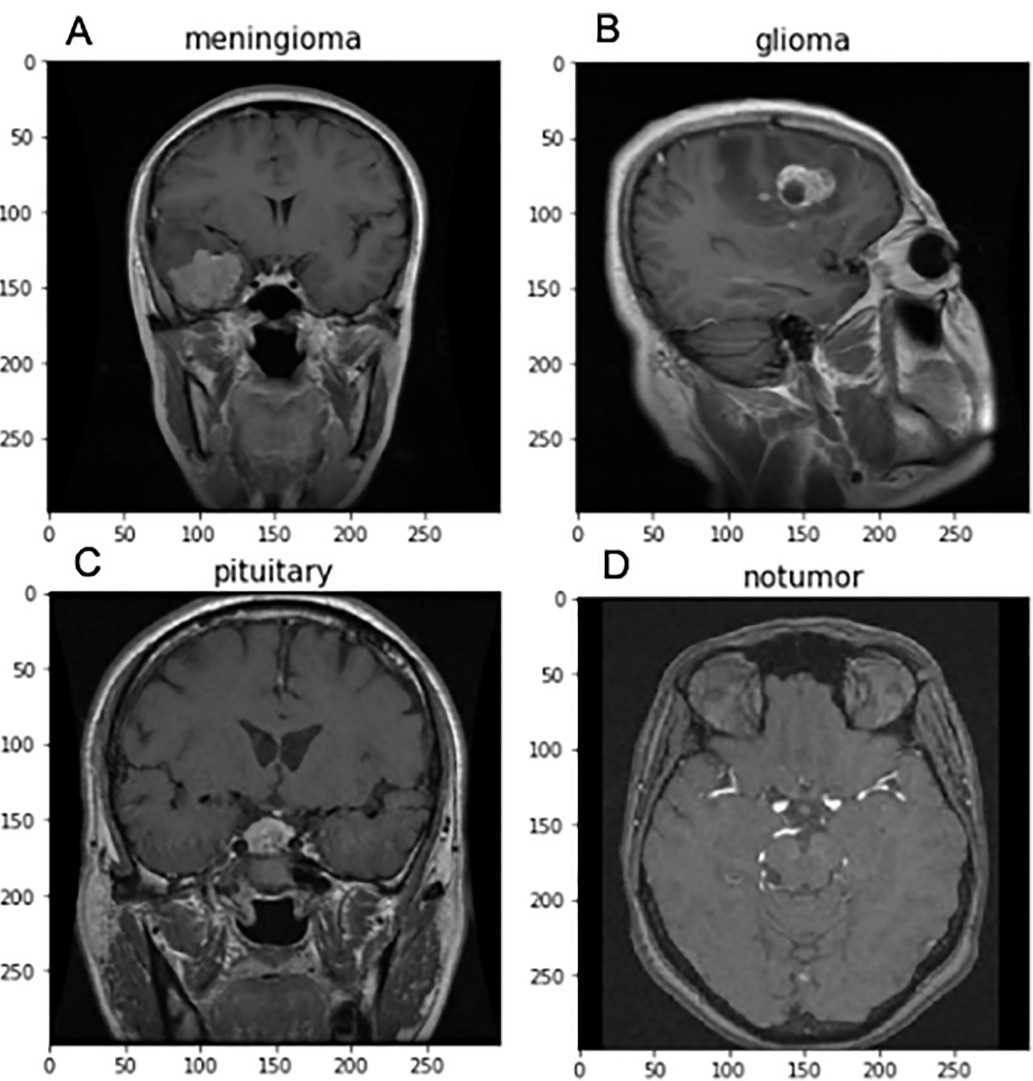

**Figure 4 Sample brain MRI images labeled by category: (A) Meningioma, (B) Glioma, (C) Pituitary, and (D) No tumor.**

and no tumor. Three datasets—Figshare, SARTAJ, and Br35H—were combined to form this dataset. A representative sample image for each class label is shown in Fig. 4. The dataset is organized into two main folders: Training and Testing. Each folder contains subfolders corresponding to the four class labels: meningioma, glioma, pituitary, and no tumor. The Training folder contains 5,712 images, while the Testing folder contains 1,311 images. The class distribution in each folder is illustrated in Fig. 5. We further split the images in the Testing folder into validation and testing sets, with 655 images for validation and 656 images for testing. We created a simulation environment with 10 clients and a central server with a global model. Each client holds a replica of the global model and acts as a validator for the previous client. The training data is split among the 10 clients, and the validation data is distributed to all clients for client evaluation.

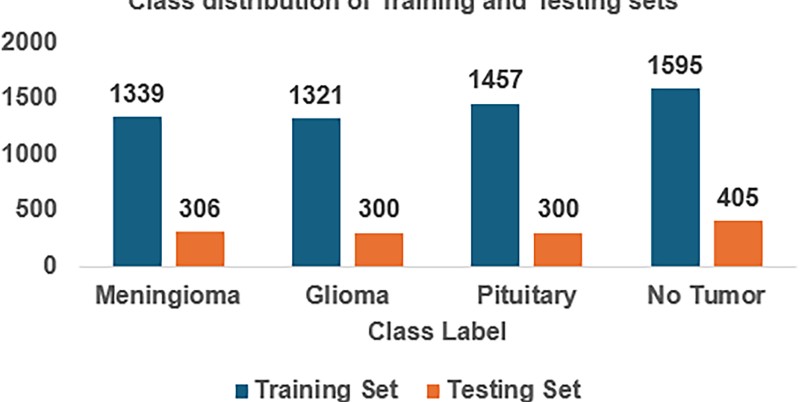

**Figure 5 Class distribution of the Kaggle brain tumor MRI dataset.**

## Experimental setup

As discussed earlier, a simulation environment is created to establish a client-server framework, consisting of a single central server and 10 clients. A common CKKS *context* is generated by a trusted CA before the FL process begins. This *context*, which implicitly contains the required public and private keys, is distributed to all clients by the CA. All clients use this shared *context* throughout the FL rounds for encryption and decryption. This design simplifies key management and ensures that encrypted model updates remain compatible for aggregation on the server. The entire FL process is implemented from scratch using PyTorch, without relying on any existing FL frameworks. For CKKS HE, the TenSEAL package is utilized. The implementation is carried out using Jupyter Notebook in Python on a DGX server with the following specifications: Nvidia RTX 3060 GPUs with 12 GB GDDR6 graphics and Intel Core i9 CPUs with 8 cores and 64 (2 × 32 GB) DDR4 RAM. All scripts are available at https://github.com/gswetha697/FedARCH.

The FL process is conducted over 20 rounds, with each client locally training a fine-tuned ResNet18 model during each round. We optimized the model by tuning several hyperparameters: a learning rate of 0.01, a momentum of 0.9, one epoch per round, a batch size of 32, cross-entropy loss as the loss function, and SGD as the optimizer.

## Evaluation metrics

We rigorously evaluated the FedARCH framework against state-of-the-art solutions using various evaluation metrics (*Singamsetty et al., 2024*). Accuracy is used to obtain the overall performance measure. Precision and recall are employed to assess the model's impact in reducing the number of false positives (FP) and false negatives (FN), respectively. The F1-score is calculated to balance both precision and recall. For brain tumor multi-class classification, it is crucial to not only reduce the number of FPs but also reduce FNs. An FP could cause unnecessary panic and lead to unnecessary treatment for patients, while an FN could overlook a potentially dangerous tumor, leading to delayed treatment and decreasing

**Table 3 Feature comparison of existing DL/FL frameworks and the proposed FedARCH framework.**

| Reference | DL | FL | Reputation | Weighted aggregation | Dynamic performance management | HE | Underperforming clients | Medical data |
|---|---|---|---|---|---|---|---|---|
| *Thiriveedhi et al. (2025)* | ✓ | × | × | × | × | × | × | ✓ |
| *Khan et al. (2022b)* | ✓ | × | × | × | × | × | × | ✓ |
| *Mathivanan et al. (2024)* | ✓ | × | × | × | × | × | × | ✓ |
| *Albalawi et al. (2024)* | ✓ | ✓ | × | × | × | × | × | ✓ |
| *Islam et al. (2023)* | ✓ | ✓ | × | × | × | × | × | ✓ |
| *Viet et al. (2023)* | ✓ | ✓ | × | × | × | × | × | ✓ |
| *Ay, Ekinci & Garip (2024)* | ✓ | ✓ | × | × | × | × | × | ✓ |
| *Bhatia & Samet (2023)* | ✓ | ✓ | × | × | × | × | ✓ | ✓ |
| *Lytvyn & Nguyen (2023)* | ✓ | ✓ | × | × | × | × | × | ✓ |
| *Fan et al. (2023)* | ✓ | ✓ | ✓ | × | ✓ | ✓ | ✓ | × |
| *Zhang et al. (2022)* | ✓ | ✓ | ✓ | ✓ | × | ✓ | × | ✓ |
| *Panigrahi, Bharti & Sharma (2023)* | ✓ | ✓ | ✓ | × | × | ✓ | × | ✓ |
| *Kang & Ahn (2023)* | ✓ | ✓ | × | × | × | × | × | × |
| *Truhn et al. (2024)* | ✓ | ✓ | × | × | × | ✓ | × | ✓ |
| *Kim et al. (2024)* | ✓ | ✓ | × | × | × | × | × | ✓ |
| *Yang et al. (2021)* | ✓ | ✓ | × | ✓ | × | × | × | ✓ |
| **FedARCH** | ✓ | ✓ | ✓ | ✓ | ✓ | ✓ | ✓ | ✓ |

patient survival rates. These metrics ensure that the FedARCH framework is thoroughly evaluated, thereby improving decision-making and patient outcomes.

## RESULTS

We compared the FedARCH framework with existing solutions, and the comparison is presented in Table 3. This table highlights the key features incorporated in the FedARCH framework that are not addressed in the existing work. The proposed FedARCH framework is compared with CL and standard FL with FedAvg, and the results are shown in Fig. 6. FedARCH performs on par with standard FL and is almost similar to CL. To further evaluate its robustness, Gaussian noise is added to some clients' data to observe the impact on the final model accuracy. We initially introduce noise to 10% of the clients and gradually increase this to 50% of the clients. Three different noise levels are considered: low (noise_level = 0.1), medium (noise_level = 0.4), and high (noise_level = 0.8). FedARCH is compared with the standard FL, and the results are illustrated in Figs. 7–9. The results demonstrate that as both the percentage of noisy clients and the level of noise in the clients' data increase, FedARCH efficiently resists the impact of noise, whereas the standard FL approach fails.

The impact of increasing the noise level on model accuracy is also considered and is illustrated in Figs. 10 and 11. With an increasing noise level and number of noisy clients, there is some impact on the FedARCH framework, as the accuracy slightly reduces from 99% to 94%. However, for the standard FL approach, there is a significant drop in

*Peer*J Computer Science

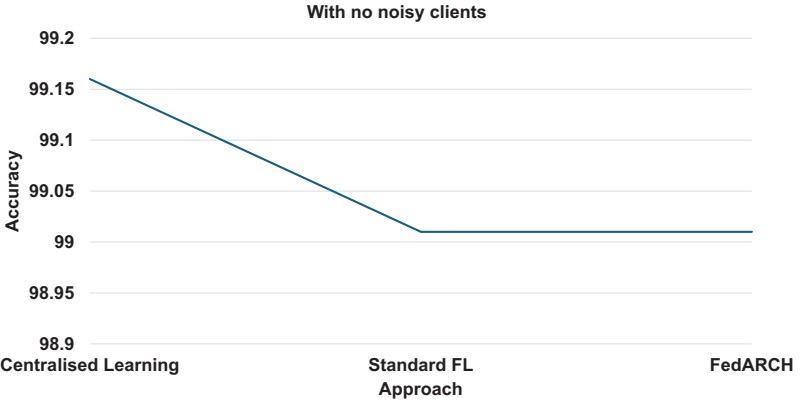

**Figure 6 Accuracy comparison of CL, standard FL, and FedARCH.**

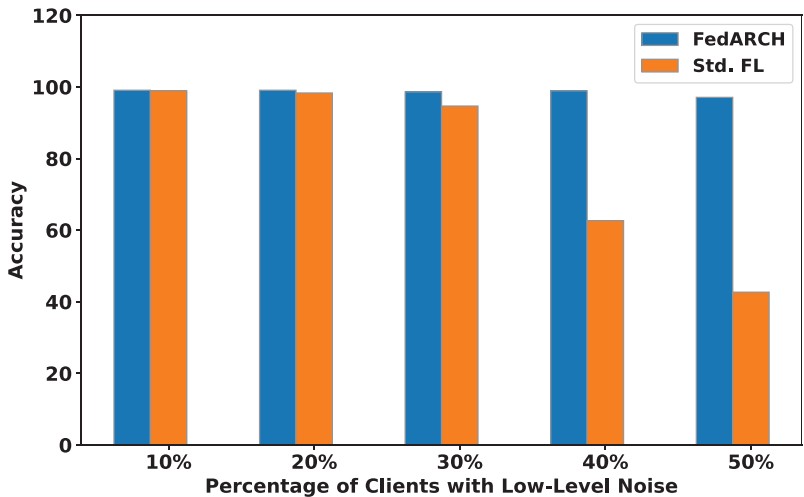

**Figure 7 Accuracy comparison of FedARCH and standard FL under varying percentages of noisy clients at a low noise level.**

performance, with accuracy plummeting from 99% to 32%. This highlights the level of resistance exhibited by our proposed FedARCH framework.

We also compare the influence of CKKS HE on both the standard and FedARCH approaches. A simulation with 40% noisy clients at a low noise level is used to evaluate the impact on both approaches, with and without CKKS. The results are shown in Fig. 12. No significant difference is observed in the FedARCH approach, but the standard approach performs better with the inclusion of CKKS. This highlights that the addition of CKKS HE does not negatively affect the performance of our model, unlike the Differential Privacy approach. This can be attributed to CKKS's ability to operate on encrypted data, real numbers, and approximate arithmetic. The accumulation of noise, which is a common issue in encryption scenarios, is effectively managed in our case. This is because we only consider plaintext-ciphertext multiplication during weighted aggregation, rather than

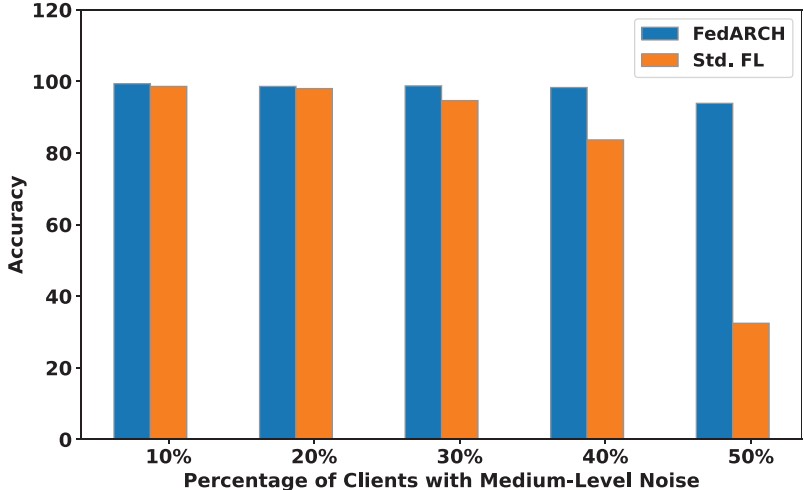

**Figure 8 Accuracy comparison of FedARCH and standard FL under varying percentages of noisy clients at a medium noise level.**

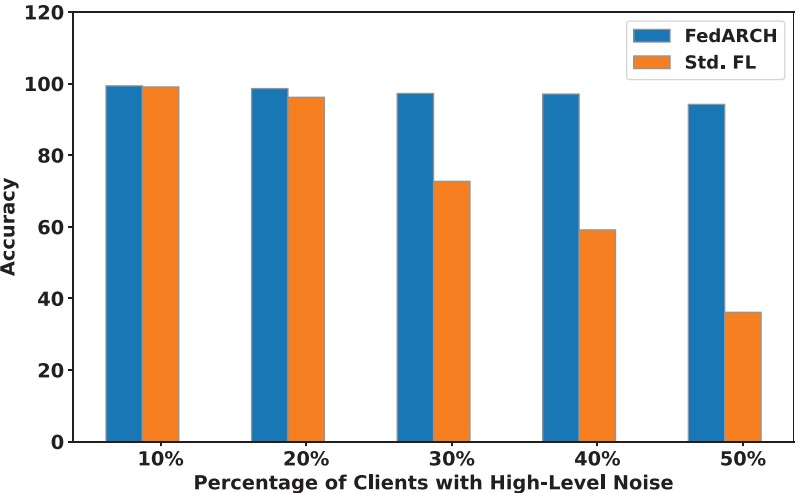

**Figure 9 Accuracy comparison of FedARCH and standard FL under varying percentages of noisy clients at a high noise level.**

ciphertext-ciphertext multiplication, which helps prevent significant noise accumulation. In this context, the plaintext refers to the normalized reputation weights, and the ciphertext refers to the encrypted local model weights.

However, the inclusion of CKKS HE incurs a slight increase in computational cost. In our work, we have primarily focused on analyzing the computational time associated with CKKS operations. Specifically, the model training without CKKS took 272 min, while using CKKS encryption extended the training time to 322 min. Although the encryption process introduces this additional computational overhead, the model's accuracy remains almost unchanged: 98.48% without CKKS and 98.32% with CKKS. It's important to note that this additional time is only associated with the training phase. Once the model is trained and

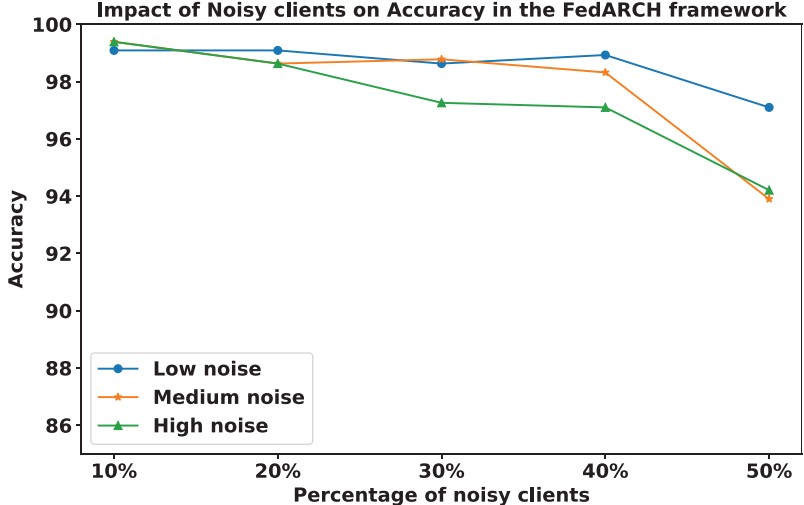

**Figure 10 Accuracy comparison of the FedARCH approach across different noise levels and varying percentages of noisy clients.**

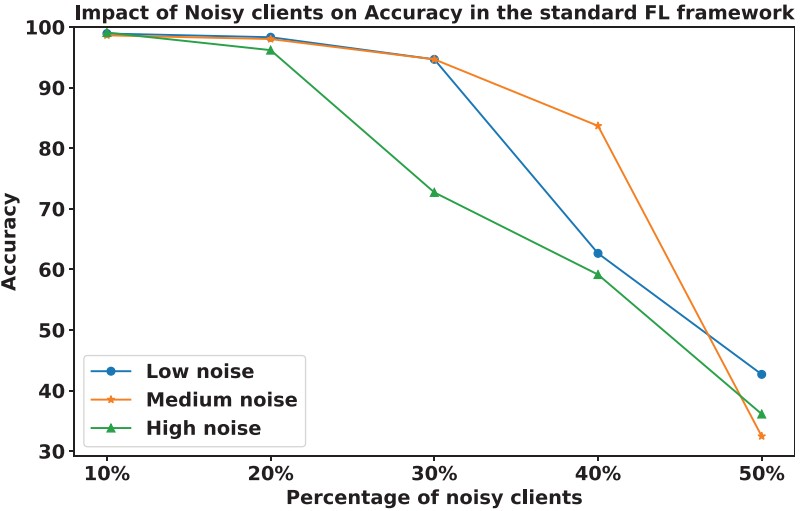

**Figure 11 Accuracy comparison of the standard FL approach across different noise levels and varying percentages of noisy clients.**

the global model is obtained, no further training is required, and the model can be used for testing. Therefore, the extra training time has no impact on the efficiency of the subsequent testing phase. The added security provided by CKKS encryption during training is a valuable trade-off, as it ensures privacy without compromising the testing performance of the model.

To address sudden spikes in performance and reduce the impact of older reputations, smoothing and decay factors are considered. Various combinations of these factors were tested and compared to assess their impact, as shown in Fig. 13. To simulate real-time changes in performance, we altered the status of an underperforming client (client 3) to a

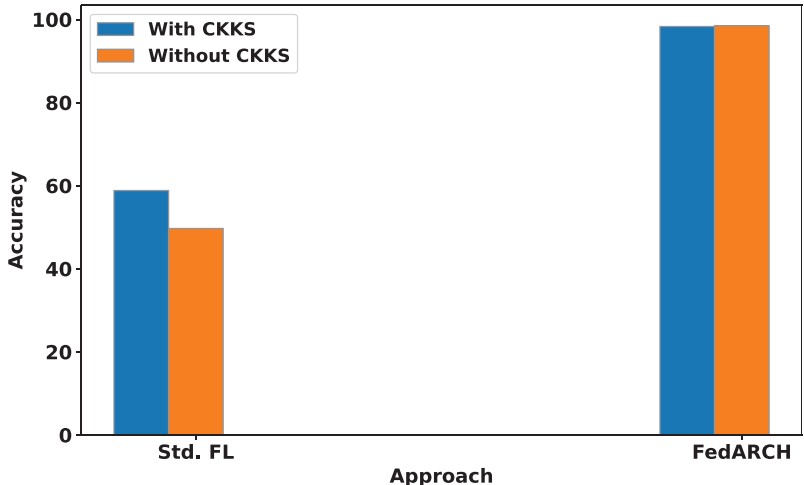

**Figure 12 Accuracy comparison of standard FL and FedARCH approaches with and without CKKS encryption, under a simulation with 40% noisy clients at a low noise level.**

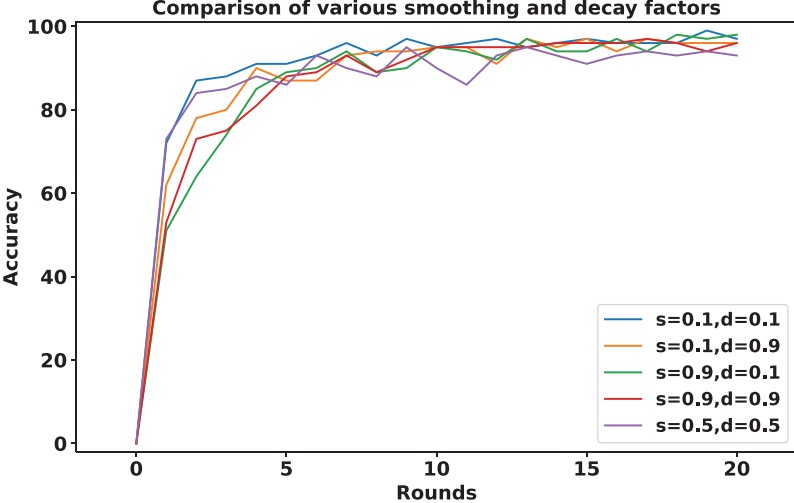

**Figure 13 Accuracy comparison for various combinations of smoothing and decay factors in the FedARCH framework.**

well-performing client and a well-performing client (client 5) to an underperforming client after round 7. Validation reports before and after this simulation are shown in Figs. 14 and 15. A rigorous evaluation was conducted using various standard metrics. Figure 16 highlights the class-wise evaluation metrics obtained by FedARCH compared to the standard FL approach. The corresponding confusion matrices are presented in Fig. 17.

## Statistical analysis

To validate the robustness and reliability of FedARCH, we conducted a comprehensive statistical analysis across 15 experimental runs. Each run involved a varying proportion of

```
Client 2 is underperforming
Client 3 is underperforming
Client 4 is underperforming
Client 6 is underperforming
```

**Figure 14 Validation report before a spike and drop simulation.**

```
Client 2 is underperforming
Client 4 is underperforming
Client 5 is underperforming
Client 6 is underperforming
```

**Figure 15 Validation report after a spike and drop simulation.**

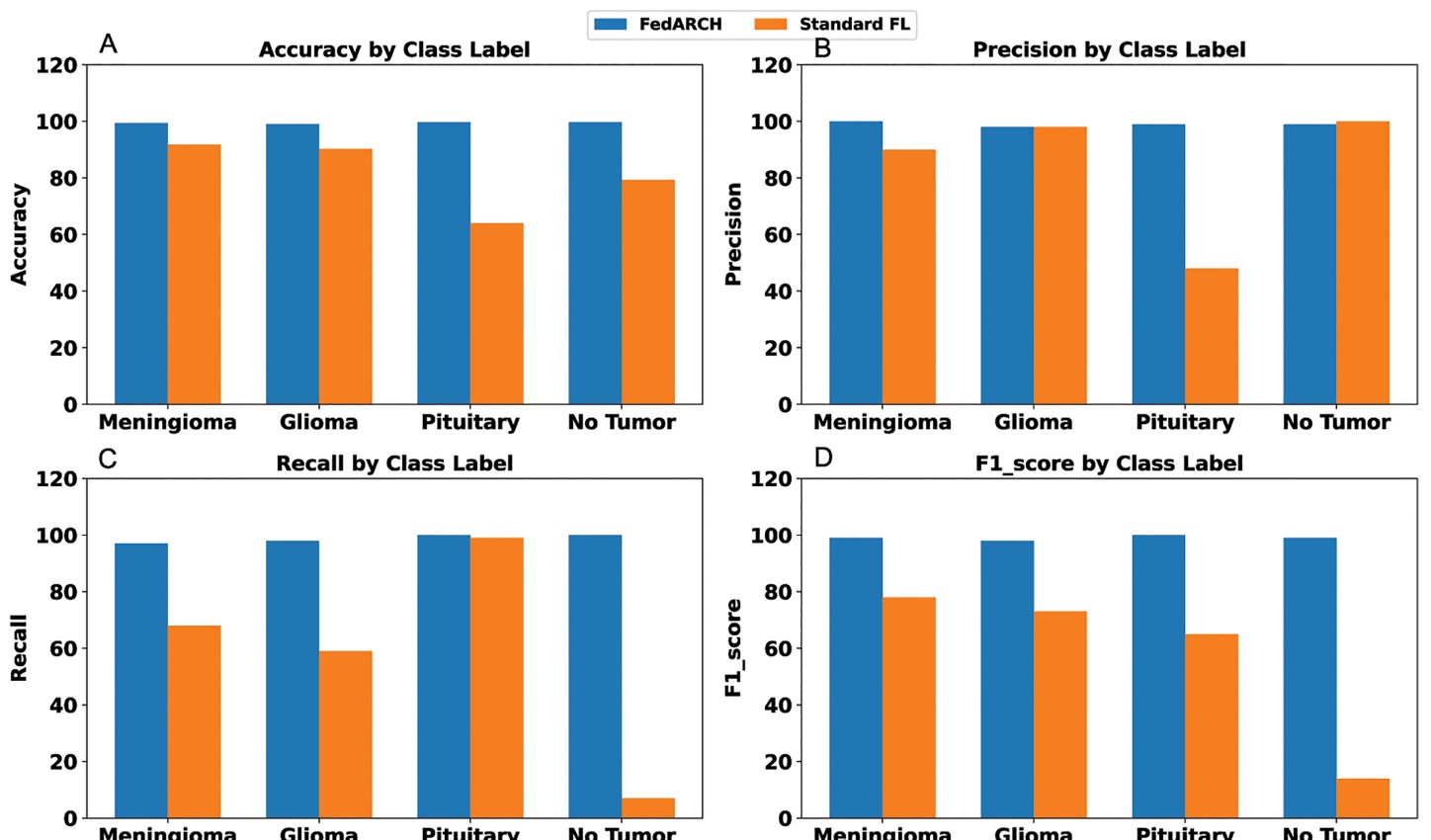

**Figure 16 Comparison of evaluation metrics for standard FL and FedARCH under a simulation with 40% noisy clients at a low noise level: (A) Accuracy by class label, (B) Precision by class label, (C) Recall by class label, and (D) F1-score by class label.**

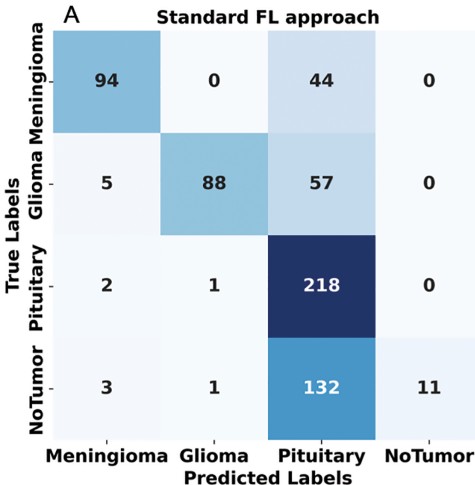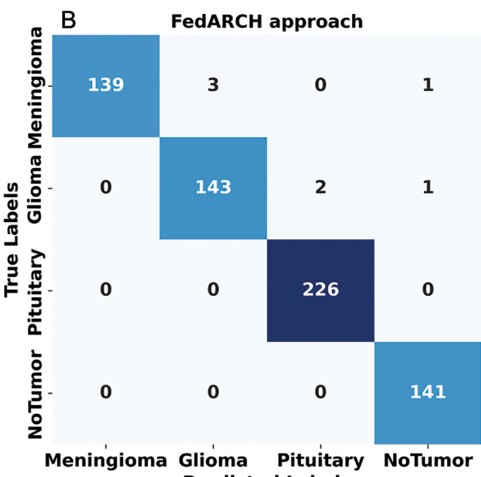

**Figure 17 Confusion matrices for standard FL and FedARCH approaches under a simulation with 40% noisy clients at a low noise level: (A) Standard FL approach, (B) FedARCH approach.**

noisy clients (10% to 50%) and different noise levels applied to the images. FedARCH achieved a mean accuracy of 97.90%, with a standard deviation of 1.74. The 95% confidence interval for the mean accuracy was [96.94%, 98.86%], indicating strong consistency and reliability across all runs. In contrast, the standard FL method exhibited significantly lower mean performance and higher variance as the proportion of noisy clients increased, with a mean accuracy of 77.87%, a standard deviation of 24.98, and a 95% confidence interval for the mean accuracy ranging from 64.04% to 91.70%. The results are illustrated in Figs. 18 and 19.

A paired t-test was performed between the accuracies of standard FL and FedARCH over the same 15 experimental conditions. The test yielded a *p*-value of 0.0052, which is below the conventional 0.05 threshold, confirming that the observed improvement is statistically significant.

## Security analysis

### Formal analysis

FedARCH is robust not only in terms of performance but also with respect to security. To demonstrate this, we utilized a Python tool called Bandit (*Roy, 2023*), which is highly effective in scanning Python code for security vulnerabilities and generating a comprehensive security report. We specifically chose Bandit because it can efficiently detect dangerous code execution commands, code injection vulnerabilities, insecure key usage, and weak cryptographic practices, issues that are particularly relevant in FL scenarios. We have also used the Scyther tool (*Egala et al., 2023*), which is popular for formal security analysis of communication protocols. It can detect several attacks like man-in-the-middle (MITM) attacks, denial-of-service (DoS) vulnerabilities, replay attacks, authentication weaknesses, and key exchange security. Given the security-sensitive nature of FL, we aimed to identify and eliminate such vulnerabilities in the FedARCH framework.

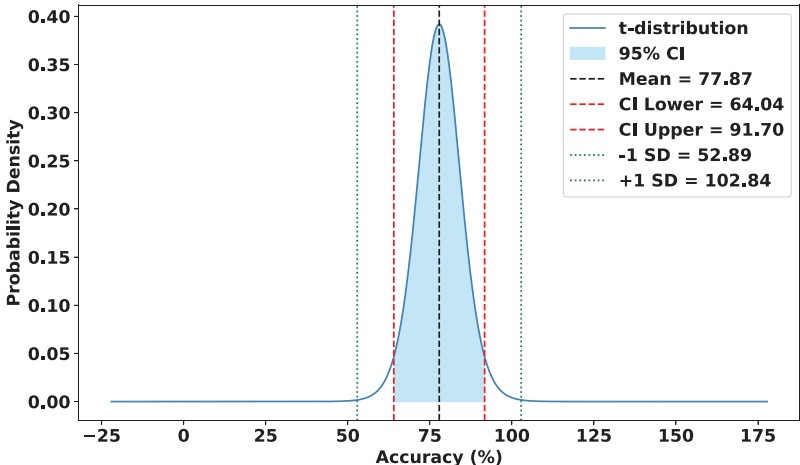

**Figure 18 Statistical analysis of the standard FL approach.**

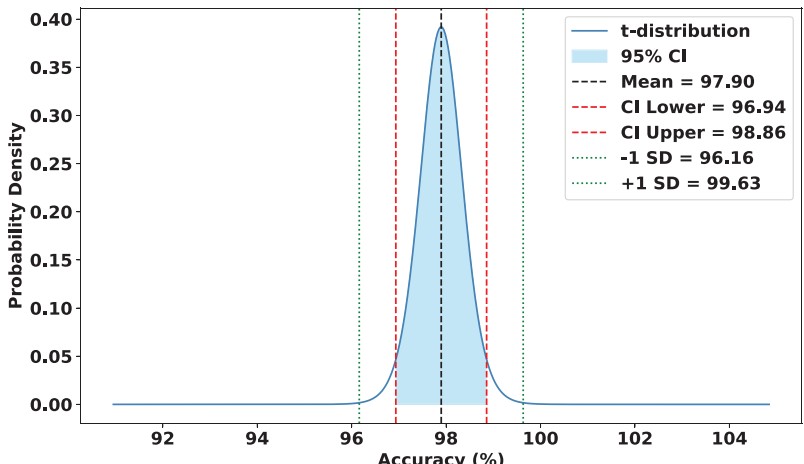

**Figure 19 Statistical analysis of the FedARCH approach.**

In our analysis, the Bandit report in Fig. 20 indicates that no issues were identified, neither by severity nor confidence, suggesting that the code adheres to security practices. The Scyther report in Fig. 21 verifies our claim and reports no attacks, confirming the robustness of FedARCH. The Bandit and Scyther reports serve as concrete evidence of FedARCH's resilience against security threats.

### Informal analysis

The CKKS HE scheme, which we considered in the FedARCH framework, facilitates the secure aggregation of encrypted weights at the server without requiring decryption in an untrustworthy environment. CKKS is based on the RLWE problem, which is NP-hard, thereby offering potential post-quantum resistance (*Lyubashevsky, Peikert & Regev, 2010*). Although clients are assumed to be trustworthy in our current implementation, the

```
[main]  INFO    profile include tests: None
[main]  INFO    profile exclude tests: None
[main]  INFO    cli include tests: None
[main]  INFO    cli exclude tests: None
[main]  INFO    running on Python 3.11.7
Run started:2025-02-15 04:19:48.214966

Test results:
        No issues identified.

Code scanned:
        Total lines of code: 174
        Total lines skipped (#nosec): 0

Run metrics:
        Total issues (by severity):
                Undefined: 0
                Low: 0
                Medium: 0
                High: 0
        Total issues (by confidence):
                Undefined: 0
                Low: 0
                Medium: 0
                High: 0
Files skipped (0):
```

**Figure 20  Bandit security analysis report for the FedARCH framework.**

reputation-aware mechanism in the FedARCH framework is designed to be robust against potential risks. It dynamically adjusts the reputation scores of clients based on validation feedback, thereby ensuring that any malicious clients attempting to poison the model with faulty updates will see their influence significantly reduced during the weighted aggregation process. Furthermore, since each client validates only one neighboring client's model, the exposure of information is limited, preventing any single client from accessing all others' updates.

# DISCUSSION

The results demonstrate that the FedARCH framework effectively mitigates the impact of underperforming clients on the final global model, whereas the standard FL approach fails as the number of noisy clients and the level of noise increase. The various evaluation metrics further validate that the FedARCH model significantly reduces false positives and

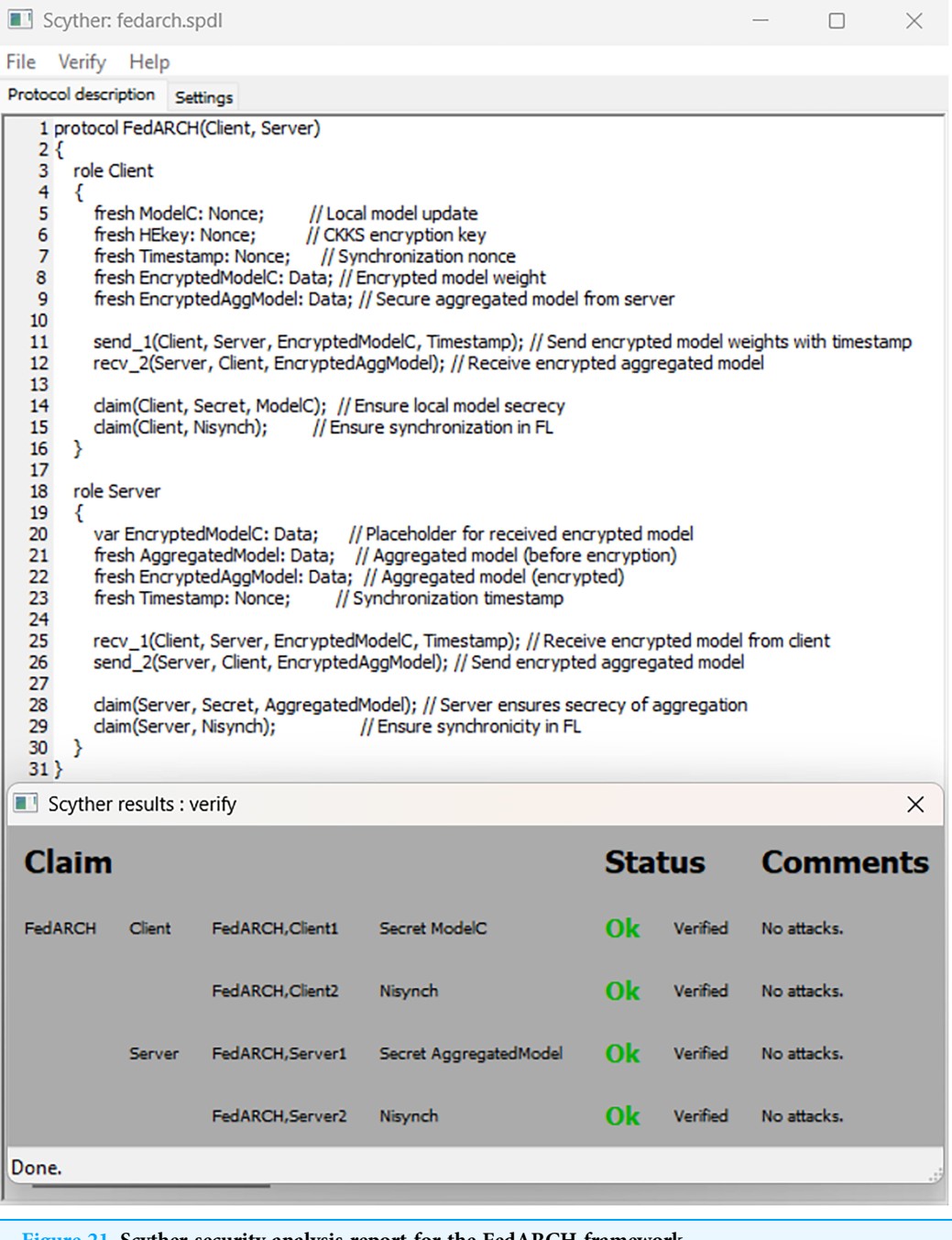

**Figure 21 Scyther security analysis report for the FedARCH framework.**

false negatives, thereby avoiding unnecessary panic and delayed treatments. Figure 22 illustrates the robustness of the proposed FedARCH framework compared to existing approaches. While *Mathivanan et al. (2024)* achieves the highest accuracy of 99.75%, it lacks the FL setup and security guarantees provided by the FedARCH framework, which achieves the next highest accuracy of 99.39%.

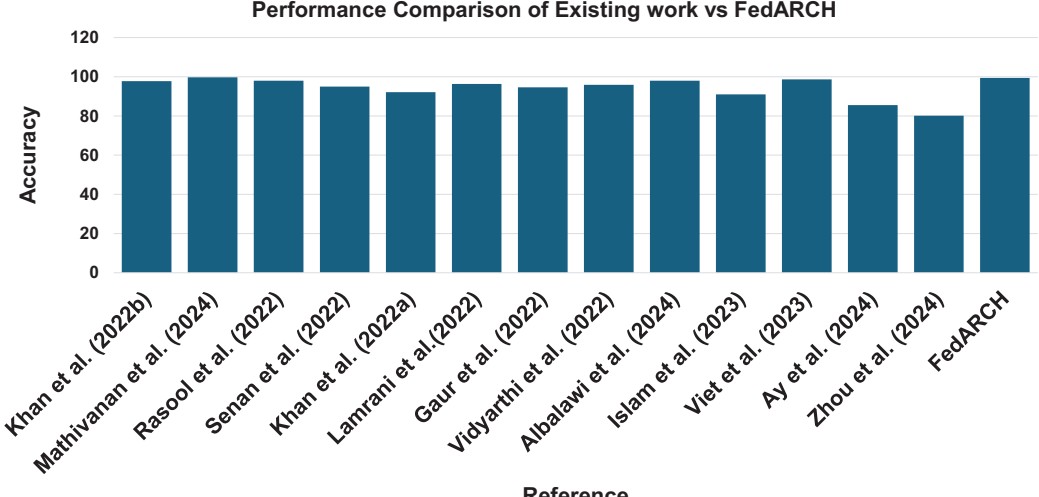

**Figure 22 Accuracy comparison of the proposed FedARCH approach with existing related work** (*Khan et al., 2022b*; *Mathivanan et al., 2024*; *Rasool et al., 2022*; *Senan et al., 2022*; *Khan et al., 2022a*; *Lamrani et al., 2022*; *Gaur et al., 2022*; *Vidyarthi et al., 2022*; *Albalawi et al., 2024*; *Islam et al., 2023*; *Viet et al. 2023*; *Ay, Ekinci & Garip, 2024*; *Zhou, Wang & Zhou, 2024*).

## CONCLUSION

In this article, we propose FedARCH, a novel FL framework that integrates reputation-aware weighted aggregation and optimized CKKS HE for brain tumor multi-classification in a cross-silo environment. Compared to state-of-the-art solutions, FedARCH not only demonstrated superior performance but also proved to be more robust in mitigating the impact of underperforming clients on the global model. In addition, underperforming clients receive feedback on their performance, enabling them to enhance their training and contribute more effectively to the collaborative learning process. This, in turn, increases prediction accuracy, ultimately facilitating better treatment options and preventive measures for patients. By integrating optimized CKKS HE, we reduce computational overhead, balancing both security and performance. The robustness of FedARCH is proved using security analysis tools like Bandit and Scyther.

## FUTURE WORK

Future work will focus on improving the applicability and robustness of FedARCH in real-world FL settings. The current implementation assumes client trustworthiness to simplify validation and aggregation using a shared CKKS HE context. While this reduces computational overhead, this assumption is a key limitation, as it may not hold in practical healthcare environments where clients can be compromised. To address this, we plan to incorporate zero-knowledge proofs and client-specific HE contexts for secure and verifiable model updates. Additionally, the framework will be validated on real-world multi-institutional medical datasets to better reflect the heterogeneity of clinical data. We also aim to extend the FedARCH approach to other medical image analysis tasks to evaluate its generalizability. In particular, we acknowledge the importance of multimodal

FL for handling diverse data types. The current work does not provide detailed processing strategies for multimodal data; thus, future efforts will focus on incorporating robust multimodal data handling capabilities into FedARCH. Finally, we will explore performance-based incentives within a blockchain-based set-up to promote honest participation and improve collaboration among clients.

### Funding
The authors received no funding for this work.

### Competing Interests
The authors declare that they have no competing interests.

### Author Contributions
- Swetha Ghanta conceived and designed the experiments, performed the experiments, analyzed the data, performed the computation work, prepared figures and/or tables, authored or reviewed drafts of the article, and approved the final draft.
- Prasanthi Boyapati conceived and designed the experiments, performed the experiments, analyzed the data, performed the computation work, prepared figures and/or tables, authored or reviewed drafts of the article, and approved the final draft.
- Sujit Biswas analyzed the data, prepared figures and/or tables, authored or reviewed drafts of the article, and approved the final draft.
- Ashok K Pradhan conceived and designed the experiments, performed the experiments, analyzed the data, performed the computation work, prepared figures and/or tables, authored or reviewed drafts of the article, and approved the final draft.
- Saraju P Mohanty analyzed the data, prepared figures and/or tables, authored or reviewed drafts of the article, and approved the final draft.

### Data Availability
The Brain Tumor MRI Dataset is available at Kaggle DOI: https://doi.org/10.34740/kaggle/dsv/2645886.
The code is available at Zenodo:
gswetha697. (2025). gswetha697/FedARCH: FedARCH v1.0.0 (FedARCH). Zenodo. https://doi.org/10.5281/zenodo.16831478.

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
