# Peer review of "Enhancing privacy-preserving brain tumor classification with adaptive reputation-aware federated learning and homomorphic encryption"

_PeerJ Computer Science, doi:10.7717/peerj-cs.3165_

## Round 0.1 · original submission · Major Revisions

**Language Note:** The review process has identified that the English language must be improved. PeerJ can provide language editing services - please contact us at [email protected] for pricing (be sure to provide your manuscript number and title). Alternatively, you should make your own arrangements to improve the language quality and provide details in your response letter. – PeerJ Staff

Reviewer 1 ·

Basic reporting

Some literature formats are inconsistent. Adjustments must be made uniformly according to the requirements of the journal. The sentence is not coherent, and the quality of the language needs to be further improved.

Experimental design

The heterogeneity of data distribution in real medical institutions may be more complex. Suggest validating FedARCH on real multi institutional medical datasets to enhance universality.

Validity of the findings

It should be improved.

Additional comments

This manuscript proposes the FedARCH framework that combines reputation-aware aggregation with efficient encryption. However, it has the following issues.
1. "Preparatory knowledge" overlaps with the introduction section.
2. Although a GitHub code repository is provided, the hyperparameter details are insufficient. Suggest adding a configuration file that includes all training parameters.
3. The manuscript did not fully discuss the computational cost of CKSS encryption.
4. Figure 19 needs to be improved.

·

Basic reporting

Thank you for submitting your manuscript. The topic of privacy-preserving brain tumor classification using federated learning is highly relevant and timely. The integration of adaptive reputation mechanisms and homomorphic encryption in the proposed FedARCH framework is noteworthy. However, to meet the scientific rigor and clarity required for publication, the following major revisions are necessary:

- Several figures (e.g., architecture diagrams, bar graphs) are crowded and lack self-contained explanations.

- Some tables duplicate content that can be summarized more effectively.

-The manuscript would benefit from thorough language polishing. Some paragraphs are overly verbose and could be streamlined for better clarity.

-Citations and references are generally appropriate but need to follow a consistent format throughout the text.

Experimental design

-The manuscript presents several valuable components (reputation-aware aggregation, smoothing and decay factors, CKKS encryption), but the novelty is not well-distinguished from prior work.

-Several existing federated learning frameworks already use client reputation scoring and encryption. It’s unclear how FedARCH uniquely combines or improves upon these methods.

-The manuscript includes a large amount of mathematical and algorithmic content, but some sections appear more descriptive than analytical.

-The validation strategy using neighboring clients is an interesting idea but needs more justification regarding trust, synchronization, and privacy.

Validity of the findings

-The experiments are conducted on a single dataset, using a simulated 10-client setup. While performance metrics are promising, this setup limits generalizability.

-There is no discussion on cross-dataset evaluation, real-world applicability, or federated deployment feasibility.

-While metrics like accuracy, precision, recall, and F1-score are reported, no statistical analysis is presented.

-Confidence intervals, standard deviations, or significance testing are absent, which limits the reliability of conclusions.

-The security analysis section includes Bandit and Scyther tools but lacks a deep interpretation of the results.

-The assumption of client honesty should be explicitly discussed as a limitation, especially in real-world federated healthcare scenarios.

Reviewer 3 ·

Basic reporting

Figures are high-quality, relevant, and well-labeled (e.g., architecture diagrams, experimental results).

Literature is well-cited with a comprehensive comparative table (Table 3 and Table 4).

All source code is available via a GitHub link, which supports reproducibility.


Some minor grammatical issues and long sentences may benefit from professional editing.

The manuscript is very long (over 20+ pages). Consider moving some detailed equations and background (e.g., on CKKS) to supplementary materials.

Experimental design

The research question is clearly defined, targeting the shortcomings of FedAvg in cross-silo FL scenarios.

The system introduces reputation-based aggregation, CKKS homomorphic encryption, and feedback mechanisms, which are novel in combination.

Experiments simulate noisy clients and varying noise levels, which is crucial for medical FL systems.

A reproducible environment is described, with PyTorch and TenSEAL for HE.

Suggestions:

Assumption of fully honest clients is a limitation; consider at least discussing malicious client scenarios or proposing countermeasures.

Expand the explanation of the trusted key-sharing authority – who manages the private key in practice?

Validity of the findings

Results are robust, especially in the presence of noisy clients (FedARCH maintains 94% vs. 33% with FedAvg).

Formal and informal security analyses are provided (Bandit and Scyther), which strengthen the privacy claims.

Class-wise metrics (Table 4) are valuable for clinical applications.

Performance comparisons with state-of-the-art methods are clearly shown.

Suggestions:

Statistical significance of performance improvements (e.g., via confidence intervals or p-values) should be reported.

More details on training epochs, learning rates, and optimizer settings would help others replicate the model training.

Additional comments

This manuscript presents a timely and relevant contribution to privacy-preserving federated learning, particularly in medical contexts. Its combined use of adaptive reputation scoring and CKKS encryption offers a practical and theoretically sound advancement. The authors’ efforts to simulate realistic client variability and to ensure system robustness are commendable.

Reviewer 4 ·

Basic reporting

The manuscript presents an innovative approach through the FedARCH framework for secure federated learning in brain tumor classification. While the study is well-structured and technically robust, a few areas need refinement to align with PeerJ’s quality standards. Some sections of the manuscript contain typographical and grammatical issues for example, the incorrect use of “clients9 data” which may affect the readability. A detailed proofreading and possible assistance from a language editing service would enhance clarity and flow.

Additionally, maintaining consistency in terminology throughout the manuscript is essential. The name “FedARCH” is not always presented uniformly, and ensuring consistent formatting will contribute to a more polished presentation. In the technical explanation of the client validation process, there is a lack of detail regarding the encryption key management. Clarifying whether a shared or rotating key approach is used for decrypting model weights would improve the transparency and reproducibility of the method.

Figures and tables, while informative, would benefit from clearer captions. For instance, brief explanations accompanying Figure 2 and Table 3 could help readers better understand their content without referring extensively back to the main text. The conclusion could also be expanded to mention possible real-world applications, such as its deployment across healthcare institutions, and outline future work such as extending the method to environments where trust cannot be assumed among participants.

By addressing these aspects, the authors can significantly improve the clarity, impact, and overall quality of the manuscript, ensuring it meets the expectations of a high-standard scientific publication.

Experimental design

The manuscript presents a strong framework, but a few issues need attention to meet PeerJ’s standards. There are minor typographical and grammatical errors (e.g., “clients9 data”) a thorough language review is advised. The term “FedARCH” should be used consistently. Clarification is needed on encryption key management during client validation. Figures like Figure 2 and Table 3 require clearer captions. The conclusion would benefit from mentioning real-world applications and future directions. Addressing these points will improve clarity and overall quality.

Validity of the findings

The manuscript is well-structured and presents a valuable contribution, but some areas require improvement. Minor typographical and grammatical errors (e.g., “clients9 data”) should be corrected, and terminology like “FedARCH” should be consistently used. Clarify the encryption key management process in the client validation mechanism. Additionally, improve the clarity of figure and table captions. Expanding the conclusion to include practical applications and future directions would further strengthen the paper.

Additional comments

no

---

## Round 0.2 · Minor Revisions

Please address the remaining requests of one of the reviewers thoroughly.

Reviewer 1 ·

Basic reporting

There are still the following issues that need further clarification in this article. Multimodal federated learning is important for multimodal data in brain tumor classification. However, this article does not provide detailed explanations and clarifications on the processing of multimodal data. This type of work can be referred to as Multimodal federated learning: Concept, methods, applications, and future directions, etc. In addition, how to mine discriminative features in uncertain data environments is an important step in classification tasks. This type of work is in the literature FFS-MCC: Fusing approximation and fuzzy uncertainty measures for feature selection with multi-correlation collaboration, Interactive and Complementary Feature Selection via Fuzzy Multigranularity Uncertainty Measures. Please further analyze this article regarding the specific explanation provided in the document. It is necessary to conduct a detailed analysis of the impact of data uncertainty and feature multiple correlations on classification performance.

Experimental design

-

Validity of the findings

-

·

Basic reporting

The author addressed all the comments thoroughly.

Experimental design

-

Validity of the findings

-

Reviewer 3 ·

Basic reporting

The author incorporated all the suggested changes.

Experimental design

-

Validity of the findings

-

Reviewer 4 ·

Basic reporting

The authors have adequately addressed all major revision points, and the revised version represents a substantial improvement. I recommend acceptance after minor revision focused on fine-tuning language and figure captions.

Experimental design

-

Validity of the findings

-

---

## Round 0.3 · accepted · Accept

Thank you for your contribution.

·

Basic reporting

All the comments are addressed.

Experimental design

All the comments are addressed.

Validity of the findings

All the comments are addressed.

Reviewer 3 ·

Basic reporting

The authors have addressed prior language-related concerns and have improved figure captions for clarity. The literature review is comprehensive, referencing current and relevant works in the field, including recent developments such as multimodal federated learning.

Experimental design

e authors have proposed a novel framework—FedARCH—integrating adaptive reputation-based aggregation and CKKS homomorphic encryption for secure federated learning. Their simulations, including the use of noisy client data, represent a rigorous test of the method’s robustness. Methods and setup are described in enough detail to allow reproducibility.

Validity of the findings

The findings are valid and supported by robust experimentation. Underlying data appears to be appropriately controlled and statistically sound. The authors present a logical flow from their methods to the conclusions.

Additional comments

The authors have adequately addressed all reviewer comments from the prior round